# Air Pollutants Interaction and Gender Difference on Bone Mineral Density T-Score in Taiwanese Adults

**DOI:** 10.3390/ijerph17249165

**Published:** 2020-12-08

**Authors:** Yu-Hsuan Lin, Chen-Feng Wang, Hsuan Chiu, Bo-Cheng Lai, Hung-Pin Tu, Pei-Yu Wu, Jiun-Chi Huang, Szu-Chia Chen

**Affiliations:** 1Department of General Medicine, Kaohsiung Medical University Hospital, Kaohsiung 807, Taiwan; a1122821m@gmail.com (Y.-H.L.); mickey990055@gmail.com (H.C.); 2Institute of Electronics, National Chiao Tung University, Hsinchu 300, Taiwan; a114n.d425y@gmail.com (C.-F.W.); bclai@mail.nctu.edu.tw (B.-C.L.); 3Department of Public Health and Environmental Medicine, School of Medicine, College of Medicine, Kaohsiung Medical University, Kaohsiung 807, Taiwan; p915013@kmu.edu.tw; 4Division of Nephrology, Department of Internal Medicine, Kaohsiung Medical University Hospital, Kaohsiung Medical University, Kaohsiung 807, Taiwan; wpuw17@gmail.com (P.-Y.W.); karajan77@gmail.com (J.-C.H.); 5Department of Internal Medicine, Kaohsiung Municipal Siaogang Hospital, Kaohsiung Medical University, Kaohsiung 812, Taiwan; 6Faculty of Medicine, College of Medicine, Kaohsiung Medical University, Kaohsiung 807, Taiwan; 7Research Center for Environmental Medicine, Kaohsiung Medical University, Kaohsiung 807, Taiwan

**Keywords:** osteoporosis, air pollutants, gender difference

## Abstract

Osteoporosis is defined as a systemic skeletal disease characterized by a reduction in bone mass and microarchitectural deterioration of bone tissue. Previous studies have reported associations between air pollution and lower bone mineral density; however, few studies have investigated the association between air pollution and osteoporosis. In this study, we combined two databases, the first including 5000 individuals registered in the Taiwan Biobank, and the second containing detailed daily data on air pollution. After multivariable adjustments, ozone (O_3_) (unstandardized coefficient β, 0.015; *p* = 0.008) was significantly positively associated with T-score, whereas carbon monoxide (CO) (unstandardized coefficient β, −0.809; *p* < 0.001), sulfur dioxide (SO_2_) (unstandardized coefficient β, −0.050; *p* = 0.005), nitric oxide (NO) (unstandardized coefficient β, −0.040; *p* < 0.001), nitrogen dioxide (NO_2_) (unstandardized coefficient β, −0.023; *p* < 0.001), and nitrogen oxide (NO_x_) (unstandardized coefficient β, −0.017; *p* < 0.001) were significantly negatively associated with T-score. The interactions between CO and NO_x_ (*p* = 0.001) and SO_2_ and NO_2_ (*p* = 0.004) on T-score were statistically significant. An increase in exposure to CO, NO and NO_x_ was associated with a faster decline in T-score in the female participants compared to the male participants. In addition, an increase in O_3_ was associated with a faster increase in T-score in the female participants compared to the male participants. In conclusion, the air pollutants CO, SO_2_, NO, NO_2_, and NO_x_ were associated with osteoporosis. In addition, there were interaction and synergetic effects between CO and NO_x_ and SO_2_ and NO_2_ on T-score. We also observed differences in the associations between air pollutants and T-score between the female and male participants.

## 1. Introduction

Osteoporosis is defined as a systemic skeletal disease characterized by a reduction in bone mass and microarchitectural deterioration of bone tissue resulting in increased bone fragility and a higher risk of fractures [1]. Fractures of the vertebrae and hip are associated with increased morbidity and mortality and are the most serious consequences of osteoporosis [2]. Lifestyle factors such as vitamin D deficiency, inadequate physical activity, low calcium intake, smoking, and alcohol abuse have been associated with an increased risk of osteoporosis and fractures [3]. Due to changes in hormone levels during menopause and aging, women have a higher prevalence of osteoporosis than men [4]. Previous studies have reported associations between air pollution and lower bone mineral density (BMD), increased BMD loss, and an increased risk of osteoporotic fractures [5,6]. However, few studies have investigated the association between air pollution and osteoporosis.

Air pollution has decreased or stabilized in many countries; however, it remains high in Taiwan [7], mostly caused by the burning of fossil fuels [8]. Increased risks of cardiovascular and respiratory morbidity and mortality have been associated with acute and chronic exposure to air pollution [9,10,11,12,13,14]. In addition, several studies have indicated that air pollution can also induce systemic and tissue-specific inflammation [15,16,17]. Recent studies have demonstrated that exposure to air pollution such as particles with an aerodynamic diameter of ≤2.5 μm (PM_2.5_), carbon monoxide (CO), sulfur dioxide (SO_2_), and nitrogen dioxide (NO_2_) is associated with arthritis [18], osteoporosis [19], reduction in BMD [5] and fractures [20,21]. Particulate matter has also been shown to cause systemic oxidative damage [22], increase in osteoclasts [23] and inflammation [24], which can result in a faster rate of bone loss and higher risk of bone fractures in older individuals. According to recent review article, potential mechanisms behind the association between outdoor air pollution, especially particulate matter, and bone damage included (1) several different atmospheric pollutants can induce low-grade systemic inflammation, which affects bone metabolism through a specific effect of cytokines such as tumor necrosis factor α (TNF-α), interleukin (IL)-1β, IL-6, and IL-17 on osteoblast and osteoclast differentiation and function; (2) some pollutants, particularly certain gas and metal compounds, can cause oxidative damage in the airway and bone cells; (3) different groups of pollutants can act as endocrine disruptors when binding to the receptors in bone cells, changing their functioning; and (4) air pollution can directly and indirectly cause vitamin D deficiency [25].

In this study, we combined two databases. The first included 5000 individuals registered in the Taiwan Biobank (TWB), and the second was the Taiwan Air Quality Monitoring Database (TAQMD), which includes detailed daily data on air pollution in Taiwan. We evaluated the correlations between air pollutants and bone marrow T-score, and explored whether there were interaction and synergetic effects between various air pollutants. Furthermore, we also explored sex differences in these associations.

## 2. Materials and Methods

### 2.1. Ethics Statement

TWB received ethical approval from the Institutional Review Board on Biomedical Science Research/IRB-BM, Academia Sinica, Taiwan, and from the Ethics and Governance Council of the TWB, Taiwan. Written informed consent was obtained from each participant in accordance with institutional requirements and the principles of the Declaration of Helsinki. In addition, this study was approved by the Institutional Review Board of Kaohsiung Medical University Hospital (KMUHIRB-E(I)-20180242).

### 2.2. The Taiwan Biobank

The TWB is the largest government-supported biobank in Taiwan, and it was established to record genomic and lifestyle data of Taiwanese residents [26,27]. The TWB includes the data of community-based volunteers aged 30 to 70 years with no history of cancers. All participants sign informed consent forms, provide blood samples and information via questionnaires in face-to-face interviews with TWB researchers and physical examinations. In this study, we included 5000 individuals registered in the TWB until April 2014.

The TWB includes data on body height and weight, and body mass index (BMI) was calculated as weight (kg)/height (m)^2^. The information obtained from the questionnaires included both personal and lifestyle factors. Regular exercise was defined as participating in leisure activities such as hiking, cycling, jogging, playing basketball, swimming, yoga, and playing console/computer-based exercise games for at least 30 min three times a week. Occupational activities such as heavy manual and physical work were not included as “exercise” in this study.

### 2.3. Collection of Demographic, Medical, and Laboratory Data

The following baseline variables were recorded: demographic features (age and sex), smoking history, medical history (diabetes mellitus (DM) and hypertension), examination findings (systolic (SBP) and diastolic blood pressures (DBP)) and laboratory data (fasting glucose, triglycerides, total cholesterol, HDL-cholesterol, LDL-cholesterol, hemoglobin, estimated glomerular filtration rate (eGFR) and uric acid). EGFR was calculated using the Modification of Diet in Renal Disease 4-variable equation [28].

### 2.4. Assessment of Bone Mineral Density

The BMD (g/cm^2^) of the calcaneus in the non-dominant foot was measured using an Achilles InSight ultrasound device (Achilles InSight, GE, Fort Myers, FL, USA). The T-score was defined as follows: (individual’s BMD—young-adult mean BMD)/standard deviation of the young-adult normal population. A T-score > −1.0 was defined as normal BMD, osteopenia as between −1.0 and −2.5, and osteoporosis as <−2.5.

### 2.5. Assessment of Air Pollutants

The TAQMD was established by the Taiwan Environmental Protection Administration, Executive Yuan and includes daily concentrations of air pollutants from 74 ambient air quality monitoring stations distributed around Taiwan. The TWB and TAQMD were linked by the area of residence of the participants and the location of the air quality monitoring stations. Outdoor air pollution exposure was estimated at the residential address of each participant. The average concentrations of air pollutants, including PM_2.5_, particles with an aerodynamic diameter of ≤10 μm (PM_10_), O_3_, CO, SO_2_, nitric oxide (NO), NO_2_, and nitrogen oxide (NO_x_) during the enrolled year were calculated. We determined average yearly data using the following three-step method: (1) Obtain the corresponding longitude and latitude of the address using Google geocoding; (2) For an interpolation point, we identified the nearest air quality monitoring station from that point; (3) From the mapped station data, we filtered the data from the survey date to the previous year, and computed the average of each air pollution metric. Four hundred and five participants without complete air pollution measurements during the enrollment period were excluded, and the remaining 4595 participants (mean age 49.7 ± 10.7 years, 2118 males) were included in this study.

### 2.6. Example of Nearest Neighbor Interpolation

To illustrate our interpolation method and dataset distribution, we plotted the locations of the monitoring stations (large circles) and participants (small circles) (Figure 1). The figure shows that we mapped the participants to the nearest station. Hence, the color of the participants is the same as the mapped station.

### 2.7. Statistical Analysis

Statistical analysis was performed using SPSS version 19.0 for Windows (SPSS Inc. Chicago, USA). Data were expressed as percentage, mean ± standard deviation, or median (25th–75th percentile) for triglycerides. Differences between groups were analyzed using the chi-square test for categorical variables and the independent t test for continuous variables. Linear regression analysis was used to identify the association between each air pollutant and BMD T-score. Interactions between air pollutants and BMD T-scores were analyzed using a generalized linear model. Figures for the synergistic effect of air pollutants on T-score were plotted using the LOESS procedure, which is a nonparametric method used to estimate regression surfaces. The LOESS procedure was performed using SAS (version 9.4, SAS Institute, Cary, NC, USA). A *p* value of less than 0.05 was considered to indicate a statistically significant difference.

## 3. Results

The mean age of the 4595 participants was 49.7 ± 10.7 years and included 2118 males and 2477 females. The participants were stratified into two groups according to a BMD T-score of either ≥−1.0 (n = 2870, 62.5%) or <−1 (n = 1725, 37.5%). A comparison of the clinical characteristics between these two groups is shown in Table 1. Compared to the participants with a T-score ≥ −1.0, those with a T-score < −1 were older, more predominantly male, had higher rates of smoking history, hypertension and regular exercise, higher SBP, DBP, fasting glucose, triglycerides, total cholesterol, LDL-cholesterol, hemoglobin and uric acid, and lower HDL-cholesterol, eGFR, and BMI. Regarding air pollutants, the participants with a T-score < −1 had lower exposure to PM_2.5_, PM_10_ and O_3_, and higher exposure to CO, NO, NO_2_, and NO_x_.

### 3.1. Correlations between Air Pollutants and BMD T-Score

We performed univariable linear regression analysis to survey the determinants of BMD T-score in the study participants. PM_2.5_ (unstandardized coefficient β, 0.007; *p* = 0.003), PM_10_ (unstandardized coefficient β, 0.007; *p* < 0.001), and O_3_ (unstandardized coefficient β, 0.016; *p* = 0.007) were significantly positively associated with T-score, whereas CO (unstandardized coefficient β, −0.841; *p* < 0.001), NO (unstandardized coefficient β, −0.043; *p* < 0.001), NO_2_ (unstandardized coefficient β, −0.017; *p* < 0.001), and NO_x_ (unstandardized coefficient β, −0.016; *p* < 0.001) were significantly negatively associated with T-score.

Table 2 shows the determinants of BMD T-score in the study participants using multivariable linear regression analysis after adjusting for age, sex, smoking history, DM, hypertension, BMI, SBP, DBP, fasting glucose, log triglycerides, total cholesterol, HDL-cholesterol, LDL-cholesterol, hemoglobin, eGFR, uric acid, and regular exercise. The results showed that O_3_ (unstandardized coefficient β, 0.015; *p* = 0.008) was significantly positively associated with T-score, and that CO (unstandardized coefficient β, −0.809; *p* < 0.001), SO_2_ (unstandardized coefficient β, −0.050; *p* = 0.005), NO (unstandardized coefficient β, −0.040; *p* < 0.001), NO_2_ (unstandardized coefficient β, −0.023; *p* < 0.001), and NO_x_ (unstandardized coefficient β, −0.017; *p* < 0.001) were significantly negative associated with T-score.

### 3.2. Interaction between Air Pollutants on BMD T-Score

Analysis of the interactions between air pollutants and BMD T-score using a generalized linear model is shown in Table 3. The interactions between CO and NO_x_ (unstandardized coefficient β, −0.026; *p* = 0.001) and SO_2_ and NO_2_ on BMD T-score (unstandardized coefficient β, −0.007; *p* = 0.004) were statistically significant. However, the interaction of other combinations did not achieve significance. Figure 2 and Figure 3 illustrate synergistic effects of CO and NO_x_ and SO_2_ and NO_2_ on BMD T-score. Synergistic effects of CO and NO_x_ and SO_2_ and NO_2_ on the association with BMD T-score were observed. This was a combined analysis trying to model air pollution effect to BMD T-score based on levels of CO and NO_X_ (Figure 2; *p* for interaction = 0.001, see Table 4) or levels of NO_2_ and SO_2_ (Figure 3; *p* for interaction = 0.004, see Table 4).

### 3.3. Correlation between Air Pollutants and BMD T-Score in Male and Female Participants

Table 4 shows the determinants of BMD T-score in the male and female participants using multivariable linear regression analysis. In the male participants, CO (unstandardized coefficient β, −0.562; *p* = 0.002), NO (unstandardized coefficient β, −0.025; *p* = 0.003), NO_2_ (unstandardized coefficient β, −0.016; *p* = 0.004), and NO_x_ (unstandardized coefficient β, −0.012; *p* = 0.001) were significantly negatively associated with T-score. In the female participants, O_3_ (unstandardized coefficient β, 0.023; *p* = 0.003) was significantly positively associated with T-score, and CO (unstandardized coefficient β, −0.870; *p* < 0.001), SO_2_ (unstandardized coefficient β, −0.074; *p* = 0.002), NO (unstandardized coefficient β, −0.043; *p* < 0.001), NO_2_ (unstandardized coefficient β, −0.026; *p* < 0.001), and NO_x_ (unstandardized coefficient β, −0.019; *p* < 0.001) were significantly negative associated with T-score. The interactions between sex and PM_2.5_ (*p* < 0.001), PM_10_ (*p* = 0.001), O_3_ (*p* = 0.021), CO (*p* = 0.033), NO (*p* = 0.007), and NO_x_ (*p* = 0.049) on T-score were statistically significant. Figure 4 illustrates the associations between the air pollutants PM_2.5_, PM_10_, O_3_, CO, SO_2_, NO, NO_2_, and NO_x_, and T-score in the male and female participants. An increase in exposure to CO, NO, and NO_x_ was associated with a slower decline in T-score in the male participants compared to the female participant. In addition, an increase in O_3_ with a faster increase in T-score in the female participants compared to the male participants.

## 4. Discussion and Conclusions

In this analysis of 5000 individuals registered in the TWB, we found evidence of a positive association between O_3_ and T-score and a negative association between other air pollutants including CO, SO_2_, NO, NO_2_, and NO_x_ and T-score. Furthermore, we found that CO and NO_x_, SO_2_ and NO_2_ had a synergetic effect on T-score. We also found differences in the associations between air pollutants and T-score between the male and female individuals.

The first important finding of this study is that we observed a positive association between O_3_ and T-score. Ozone is the main photochemical component of polluted air, and it has been shown to cause dose-dependent oxidative stress by producing free radicals through cell membrane lipoperoxidation, protein oxidation, the inactivation of enzymes, destruction of DNA, and ultimately cell apoptosis [29]. However, O_3_ therapy has increasingly been used for the treatment of herniated discs, jaw osteonecrosis, and pain management [30,31], and it has been shown to enhance complete healing of bisphosphonate-induced osteonecrosis of the jaw through the restoration of normal bone physiology [30]. An animal study of rat calvarial defects demonstrated that O_3_ had a positive effect on bone formation [32]. In addition, a randomized study of 48 rats in which topical ozone was injected at the premaxillary suture indicated that O_3_ therapy can increase osteoclasts and osteoblasts, and also stimulate bone regeneration [33]. These studies indicate that ozone may be a protective factor against osteoporosis.

In this study, CO was negatively associated with T-score, indicating that CO may be a risk factor for osteoporosis. Environmental CO is as a common component of air pollution, cigarette, and wood smoke, and its systemic toxicity is well known [34,35]. A previous study reported that an increase in exposure to CO was associated with an increase in the prevalence of osteoporosis from 13.58 to 22.25 per 1000 person–years [19]. The binding affinity for CO is high for many ferrous heme-containing proteins, and so CO causes hypoxia by reducing the capacity to carry oxygen and reducing the release of O_2_ into tissues [36]. Hypoxia has been shown to reduce the growth of osteoblasts, thereby leading to bone thinning and osteoporosis [37]. Chang et al. reported an association between exposure to CO and NO_2_ and osteoporosis [19]; however, a recent in vivo study demonstrated that CO suppresses osteoclast differentiation by inhibiting the activation of PPAR-γ induced by RANKL [38]. Therefore, further studies are warranted to explore the role of CO in osteoporosis.

Another important finding of the present study is that SO_2_ was negatively associated with T-score. SO_2_ is a common air pollutant that is toxic to various organs. These toxic effects have been studied extensively, and include oxidative damage, DNA damage and inflammation [39]. Several studies have reported significantly increased levels of serum sulfite (SO_2_ derivative) in patients with acute pneumonia and chronic renal failure, suggesting that sulfite may be a mediator of inflammatory [40,41]. Increases in the levels of the pro-inflammatory cytokines IL-6 and TNF-αhave been reported in the lungs of mice exposed to SO_2_ [42]. IL-6 has been proposed to contribute to both localized and systemic osteoclast-mediated bone destruction associated with chronic inflammation [43]. In the current study, NO, NO_2_, and NO_x_ were also negatively associated with T-score, suggesting that they may be potential risk factors for osteoporosis. The induction of NO production in bone by proinflammatory cytokines suggests that NO may act as a mediator of bone disease in patients with cytokine-associated disorders such as postmenopausal osteoporosis, tumor-associated osteolysis and rheumatoid arthritis [44]. NO has dose-dependent and biphasic effects on the skeletal system, with low concentrations (due to eNOS activity) being associated with osteocyte and osteoblast activity to mediate osteoclast bone resorption. Higher concentrations (due to iNOS activation or exogenous administration of higher doses of NO donors) may lead to bone loss [45]. An ecological retrospective cohort study based on data obtained from three databases demonstrated that NO_2_ and SO_2_ exposure was associated with hospital admissions due to hip fractures [21]. In addition, a population-based retrospective cohort study that evaluated the risk of osteoporosis in Taiwanese residents showed that NO_2_ exposure was associated with an increased risk of osteoporosis [19]. Moreover, a 1 μg/m^3^ increase in NO_2_ has been associated with a 16.5% elevated risk of osteoporosis [46]. Furthermore, a cross-sectional analysis of middle-aged adults demonstrated serum IL-6 responses to pollution-related NO_2_ exposure, whereas such responses were not seen for other pro-inflammatory cytokines including IL-8 and TNF-α [47]. Inflammatory processes can be affected by SO_2_, NO, NO_2_, and NO_x_, and they can harm the skeleton [48]. Taken together, we suggest that SO_2_, NO, NO_2_, and NO_x_ cause osteoporosis through inflammation, and thus they may be potential risk factors for osteoporosis.

We also found interactions and synergetic effects between CO and NO_x_ and SO_2_ and NO_2_ on BMD T-score. At the molecular level, the toxic effects of NO_x_ and CO have been shown with regards to hemoglobin binding, which then reduces the efficiency of O_2_ transport and the reversible (NO) or irreversible (CO) inhibition of mitochondrial oxidative phosphorylation by reversible binding to the heme aa3 site of cytochrome c oxidase [49]. Low concentrations of NO have been shown to specifically and reversibly inhibit cytochrome c oxidase in competition with oxygen in several tissues and cells in culture [50]. Therefore, we hypothesize that NO_x_ can enhance the hypoxia caused by CO and worsen osteoporosis. NO_2_ can promote the formation of sulfate, which due to hygroscopicity can create an aqueous layer over particles of mineral oxides, thereby leading to further adsorption and reaction of other pollutants under ambient atmospheric conditions, including SO_2_ [51]. As a result, we hypothesize that the synergistic effect of SO_2_ and NO_2_ may be a risk factor for osteoporosis due to the further adsorption of other pollutants.

The last important finding of this study is that increased exposure to the air pollutants CO, NO, and NO_x_ resulted in a faster decline in T-score in the female participants compared to the male participants. In addition, increased exposure to O_3_ resulted in a faster increase in T-score in the female participants comparted to the male participants. These findings indicate that there are differences in the associations between air pollutants and T-score between males and females. Hormone factors also significantly influence the loss of bone mass. Estrogen regulates RANKL, and this is mediated by the production of cytokines such as IL-1, IL-6, and TNF [52]. Therefore, we hypothesize that women may be more vulnerable than men to develop osteoporosis upon exposure to air pollution.

In this study, osteoporosis was confirmed using an Achilles InSight ultrasound device instead of dual energy X-ray absorptiometry (DXA). The Achilles InSight has been shown to be able to identify osteoporosis defined by axial BMD using DXA in Chinese women [53]. The QUS t-score at the left heel was positively correlated with the DXA t-score at the right heel (r = 0.90, *p* < 0.001) in eighty women, aged 53–73 years, with osteoporosis and/or fractures were followed repeatedly for 7 years [54]. In addition, QUS may be an improved predictor of fractures in comparison with DXA [55,56,57,58]. Although, the most widely validated technique to measure BMD is DXA, and diagnostic criteria based on the T-score for BMD are a recommended entry criterion for the development of pharmaceutical interventions in osteoporosis. However, quantitative ultrasound has several advantages over DXA, including that radiation is not required, the low cost and portability.

There are several limitations to this study. First, this is a cross-sectional study, and therefore we could not evaluate the association between air pollution and bone loss over time or the incidence of osteoporotic fractures. Follow-up studies are needed to confirm our results. Second, the estimates of air pollutants at the participants’ home addresses are crude measures of exposure to air pollution. Although outdoor air pollution is a substantial contributor to indoor/personal exposure, it may not accurately reflect actual personal exposure, and we had no additional information about air quality indoors. Third, further studies are needed to assess eventually the correlation between QUS and the gold standard technique for diagnosing osteoporosis, based on DXA or Quantitative computed tomography (QCT). Finally, it takes enough time for air pollution to cause osteoporosis. However, we don’t have the resident time in the area or where they stayed before in these participants. Therefore, we use the average concentrations of air pollutants during the enrolled year as representative. Further study is needed to confirm the relationship between exposure time and osteoporosis.

In summary, we found that exposure to air pollutants such as CO, SO_2_, NO, NO_2_, and NO_x_ was associated with osteoporosis, and O_3_ may be a protective factor against osteoporosis. In addition, there were interaction and synergetic effects between CO and NO_x_ and SO_2_ and NO_2_ on BMD T-score. Furthermore, women were more vulnerable than men to develop osteoporosis upon exposure to air pollution. Despite the research limitations, our findings highlight the importance of air pollution on osteoporosis.

## Figures and Tables

**Figure 1 ijerph-17-09165-f001:**
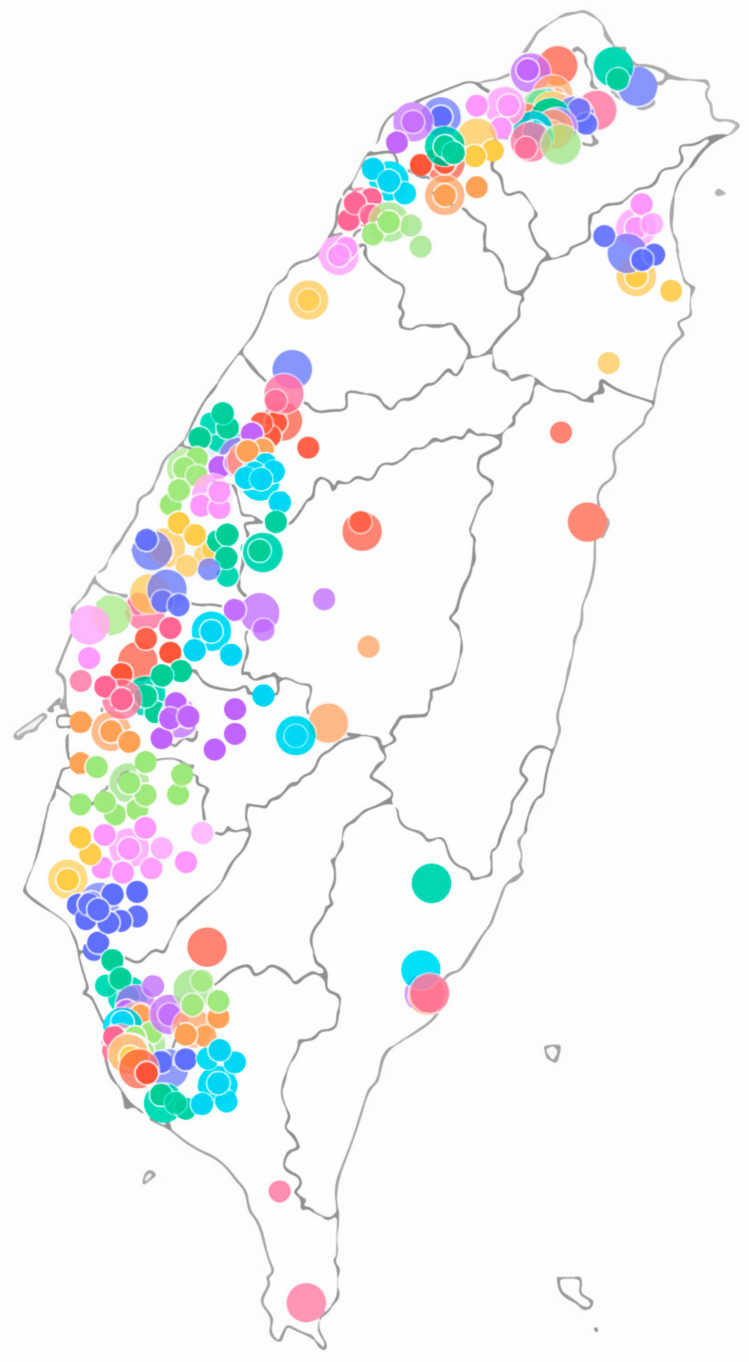
Example of nearest neighbor interpolation: Big circles stand for monitoring station, small ones for participants.

**Figure 2 ijerph-17-09165-f002:**
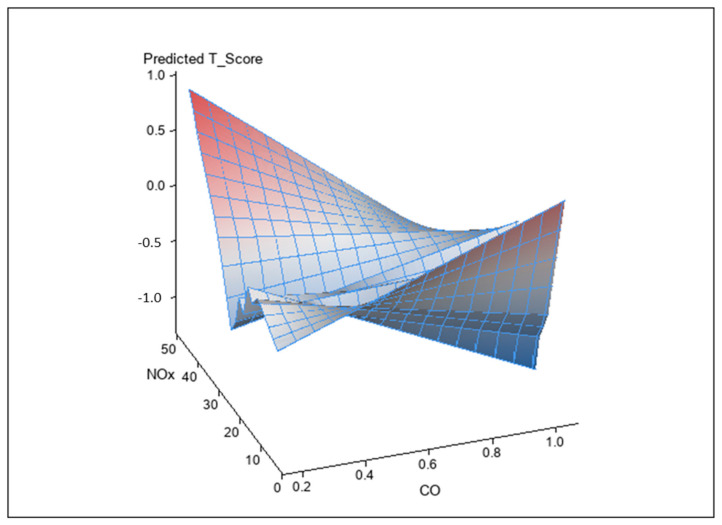
Synergistic effect of CO and NO_x_ on BMD T-score. The interaction between CO and NO_x_ on BMD T-score was statistically significant (*p* = 0.001).

**Figure 3 ijerph-17-09165-f003:**
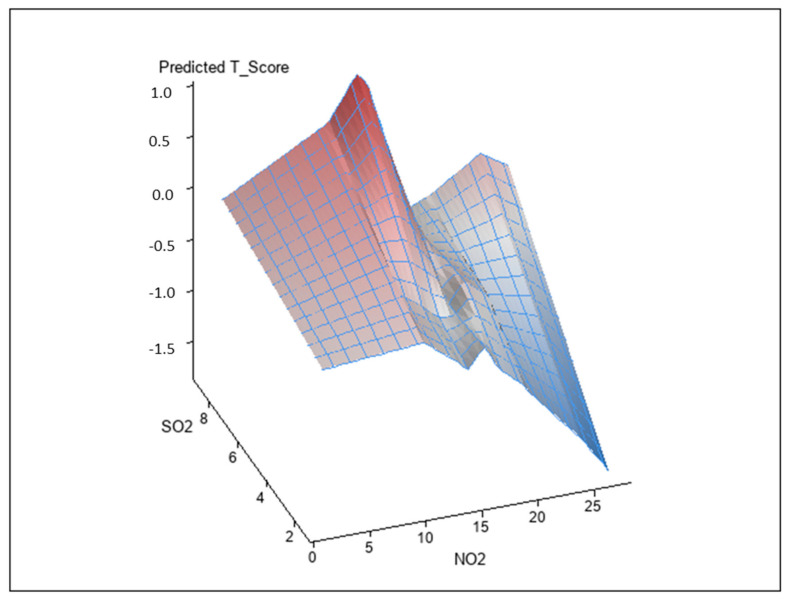
Synergistic effect of SO_2_ and NO_2_ on BMD T-score. The interaction between SO_2_ and NO_2_ on BMD T-score was statistically significant (*p* = 0.004).

**Figure 4 ijerph-17-09165-f004:**
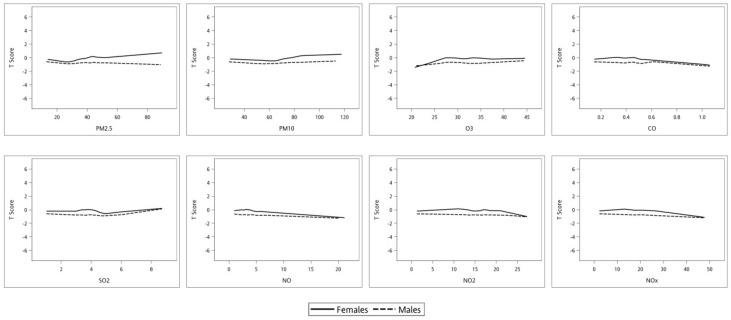
Association of air pollutants (A) PM_2.5_, (B) PM_10_, (C) O_3_, (D) CO, (E) SO_2_, (F) NO, (G) NO_2_, (H) NO_x_ with BMD T-score in male and female participants. The interaction between sex and PM_2.5_ (*p* < 0.001), PM_10_ (*p* = 0.001), O_3_ (*p* = 0.021), CO (*p* = 0.033), NO (*p* = 0.007), and NO_x_ (*p* = 0.049), in the value of T-score were statistically significant.

**Table 1 ijerph-17-09165-t001:** Comparison of clinical characteristics among participants according to bone mineral density (BMD) T score ≥ −1.0 or < −1.

Characteristics	All (n = 4595)	T-Score ≥ −1.0 (n = 2870)	T-Score < −1 (n = 1725)	*p*
Age (year)	49.7 ± 10.7	47.6 ± 10.3	53.2 ± 10.5	<0.001
Male gender (%)	46.1	40.7	54.9	<0.001
Smoking history (%)	27.1	23.8	32.6	<0.001
DM (%)	5.0	4.6	5.6	0.160
Hypertension (%)	11.1	9.2	14.1	<0.001
BMI (kg/m^2^)	24.2 ± 3.5	24.3 ± 3.5	24.1 ± 3.5	0.023
SBP (mmHg)	115.6 ± 17.1	114.1 ± 17.1	118.2 ± 17.0	<0.001
DBP (mmHg)	71.5 ± 11.1	71.0 ± 10.9	72.3 ± 11.5	<0.001
Laboratory parameters				
Fasting glucose (mg/dL)	96.7 ± 19.7	95.6 ± 18.2	98.5 ± 21.8	<0.001
Triglyceride (mg/dL)	97 (69–140)	94 (66–136)	100 (72–145)	<0.001
Total cholesterol (mg/dL)	195.5 ± 35.8	193.0 ± 35.4	199.7 ± 36.2	<0.001
HDL-cholesterol (mg/dL)	54.3 ± 13.4	54.8 ± 13.5	53.6 ± 13.3	0.003
LDL-cholesterol (mg/dL)	122.5 ± 32.2	120.6 ± 31.6	125.6 ± 33.0	<0.001
Hemoglobin (g/dL)	14.0 ± 1.6	13.9 ± 1.6	14.2 ± 1.5	<0.001
eGFR (mL/min/1.73 m^2^)	108.0 ± 25.1	109.0 ± 25.0	106.2 ± 25.1	<0.001
Uric acid (mg/dL)	5.6 ± 1.5	5.5 ± 1.5	5.7 ± 1.5	<0.001
Regular exercise habits (%)	45.2	43.1	48.6	<0.001
Air pollutants				
PM_2.5_ (μg/m^3^)	37.7 ± 10.8	38.0 ± 10.7	37.0 ± 10.8	0.003
PM_10_ (μg/m^3^)	67.8 ± 17.0	68.6 ± 17.3	66.5 ± 16.3	<0.001
O_3_ (ppb)	30.9 ± 3.9	31.0 ± 3.7	30.7 ± 4.1	0.044
CO (ppm)	0.45 ± 0.18	0.44 ± 0.16	0.47 ± 0.21	<0.001
SO_2_ (ppb)	3.7 ± 1.2	3.7 ± 1.2	3.7 ± 1.2	0.479
NO (ppb)	4.2 ± 4.1	3.9 ± 3.5	4.7 ± 4.8	<0.001
NO_2_ (ppb)	15.1 ± 5.7	14.8 ± 5.5	15.4 ± 5.9	0.001
NO_x_ (ppb)	19.3 ± 9.0	18.7 ± 8.2	20.2 ± 10.1	<0.001

Abbreviations. BMD, bone marrow density; DM, diabetes mellitus; BMI, body mass index; SBP, systolic blood pressure; DBP, diastolic blood pressure; HDL, high-density lipoprotein; LDL, low-density lipoprotein; eGFR, estimated glomerular filtration rate; PM_2.5_, particle with aerodynamic diameter of 2.5 μm or less; PM_10_, particle with aerodynamic diameter of 10 μm or less; O_3_, ozone; CO, carbon monoxide; SO_2_; sulfur dioxide; NO, nitric oxide; NO_2_, nitrogen dioxide; NO_x_, nitrogen oxide.

**Table 2 ijerph-17-09165-t002:** Association of air pollutants with BMD T-score using multivariable linear regression analysis.

Air Pollutants	Multivariable	
	Unstandardized Coefficient β (95% CI)	*p*
PM_2.5_ (per 1 μg/m^3^)	−0.002 (−0.006, 0.002)	0.311
PM_10_ (per 1 μg/m^3^)	0.001 (−0.002, 0.004)	0.491
O_3_ (per 1 ppb)	0.015 (0.004, 0.026)	0.008
CO (per 1 ppm)	−0.809 (−1.043, −0.576)	<0.001
SO_2_ (per 1 ppb)	−0.050 (−0.085, −0.015)	0.005
NO (per 1 ppb)	−0.040 (−0.050, −0.029)	<0.001
NO_2_ (per 1 ppb)	−0.023 (−0.030, −0.015)	<0.001
NO_x_ (per 1 ppb)	−0.017 (−0.022, 00.012)	<0.001

Values expressed as unstandardized coefficient β and 95% confidence interval (CI). Abbreviations are the same as in Table 1. Multivariable model: adjusted for age, sex, smoking history, DM, hypertension, BMI, SBP, DBP, fasting glucose, log triglyceride, total cholesterol, HDL-cholesterol, LDL-cholesterol, hemoglobin, eGFR, uric acid, and regular exercise.

**Table 3 ijerph-17-09165-t003:** The interaction between air pollutants on BMD T-score using generalized linear model.

Air Pollutants	Interaction	
	Unstandardized Coefficient β (95% CI)	*p*
CO × SO_2_	−0.041 (−0.229, 0.147)	0.670
CO × NO	−0.036 (−0.097, 0.025)	0.251
CO × NO_2_	−0.004 (−0.055, 0.047)	0.880
CO × NO_x_	−0.026 (−0.043, −0.010)	0.001
SO_2_ × NO	0.006 (−0.008, 0.021)	0.385
SO_2_ × NO_2_	−0.007 (−0.012, −0.002)	0.004
SO_2_ × NO_x_	−0.002 (−0.005, 0.001)	0.181
NO × NO_2_	0.002 (−0.003, 0.006)	0.447
NO × NO_x_	0.000 (−0.002, 0.002)	0.795
NO_2_ × NO_x_	0.001 (0.000, 0.002)	0.146

Values expressed as unstandardized coefficient β and 95% confidence interval (CI). Abbreviations are the same as in Table 1.

**Table 4 ijerph-17-09165-t004:** Association of air pollutants with BMD T-score using multivariable linear regression analysis in male and female participants.

Parameters	Male		Female		Interaction *p*
	Unstandardized Coefficient β (95% CI)	*p*	Unstandardized Coefficient β (95% CI)	*p*	
PM_2.5_ (per 1 μg/m^3^)	−0.005 (−0.011, 0.000)	0.060	−0.001 (−0.007, 0.005)	0.689	<0.001
PM_10_ (per 1 μg/m^3^)	0.000 (−0.004, 0.003)	0.773	0.001 (−0.002, 0.005)	0.525	0.001
O_3_ (per 1 ppb)	0.002 (−0.014, 0.017)	0.832	0.023 (0.008, 0.038)	0.003	0.021
CO (per 1 ppm)	−0.562 (−0.916, −0.207)	0.002	−0.870 (−1.175, −0.565)	<0.001	0.033
SO_2_ (per 1 ppb)	−0.016 (−0.066, 0.034)	0.522	−0.074 (−0.121, −0.027)	0.002	0.498
NO (per 1 ppb)	−0.025 (−0.042, −0.009)	0.003	−0.043 (−0.057, −0.030)	<0.001	0.007
NO_2_ (per 1 ppb)	−0.016 (−0.027, −0.005)	0.004	−0.026 (−0.036, −0.016)	<0.001	0.254
NO_x_ (per 1 ppb)	−0.012 (−0.019, −0.005)	0.001	−0.019 (−0.025, −0.013)	<0.001	0.049

Values expressed as unstandardized coefficient β and 95% confidence interval (CI). Abbreviations are the same as in Table 1. Multivariable model: adjusted for age, smoking history, DM, hypertension, BMI, SBP, DBP, fasting glucose, log triglyceride, total cholesterol, HDL-cholesterol, LDL-cholesterol, hemoglobin, eGFR, uric acid and regular exercise.

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
