# Peer review of "Air Pollutants Interaction and Gender Difference on Bone Mineral Density T-Score in Taiwanese Adults"

_ijerph, 2020, doi:10.3390/ijerph17249165_

Round 1

Reviewer 1 Report

Brief summary

Authors combined two databases to see the air pollutants interaction and gender difference on bone mineral density in Taiwanese adults. And concluded that the air pollutants CO, SO2, NO, NO2 and NOx were associated with osteoporosis.

Comments

Materials and Methods

  1. The BMD was measured using an Achilles InSight ultrasound device. Can it be used as definite diagnosis? Please authors afford more evidences.
  2. Resident with osteoporosis have to expose to air pollutants for enough period of time. Authors have to write in more details about the collected time period between air pollutants and resident profile.

Results

  1. The definition of osteoporosis is T-score less than -2.5 but in this manuscript, authors defined the BMD T-score only less than -0.1 (osteopenia). Please explained this point.

Author Response

Brief summary

Authors combined two databases to see the air pollutants interaction and gender difference on bone mineral density in Taiwanese adults. And concluded that the air pollutants CO, SO2, NO, NO2 and NOx were associated with osteoporosis.

Comments

Materials and Methods

  1. The BMD was measured using an Achilles InSight ultrasound device. Can it be used as definite diagnosis? Please authors afford more evidences.
  2. Ans: Thank you for your comments. We have added one paragraph in the Discussion.
  • In this study, osteoporosis was confirmed using an Achilles InSight ultrasound device instead of dual energy X-ray absorptiometry (DXA). The Achilles InSight has been shown to be able to identify osteoporosis defined by axial BMD using DXA in Chinese women [53]. The QUS t-score at the left heel was positively correlated with the DXA t-score at the right heel (r = 0.90, p < 0.001) in eighty women, aged 53–73 years, with osteoporosis and/or fractures were followed repeatedly during 7 years [54]. Besides, QUS may be an improved predictor of fractures in comparison with DXA [55,56,57,58]. Although, the most widely validated technique to measure BMD is DXA, and diagnostic criteria based on the T-score for BMD are a recommended entry criterion for the development of pharmaceutical interventions in osteoporosis. However, quantitative ultrasound has several advantages over DXA, including that radiation is not required, the low cost and portability. (Line 312-322)
  •  

  1. Resident with osteoporosis have to expose to air pollutants for enough period of time. Authors have to write in more details about the collected time period between air pollutants and resident profile.
  2. Ans: Thank you for your comments. We totally agreed your point about exposure time. However, we don’t have the resident time in the area or where they stayed before in these participants. We collected the average concentrations of air pollutants during the enrolled year (write in the method). We have added the issue in the limitation.
  • Finally, it takes enough time for air pollution to cause osteoporosis. However, we don’t have the resident time in the area or where they stayed before in these participants. Therefore, we use the average concentrations of air pollutants during the enrolled year as representative. Further study is needed to confirm the relationship between exposure time and osteoporosis. (Line 333-337)
  •  

Results

  1. The definition of osteoporosis is T-score less than -2.5 but in this manuscript, authors defined the BMD T-score only less than -0.1 (osteopenia). Please explained this point. 
  2. Ans: Thank you for your comments. Indeed, osteoporosis is defined as T-score less than -2.5. We show T-score less than -1 in table 1. The prevalence of T-score < -2.5 is 372/4595 (8.1%), no significant statistical difference is noted between the 2 groups. Therefore, we use T-score < -1 in table 1 to show the results. If the reviewer feels it would be suitable to show 2 groups using T-score less than -2.5, we can change the table.

Reviewer 2 Report

Dear Authors,

despite the novelty and scientific soundness of your research, many issues are hidden in the overall sufficient contribution proposal.

First of all, when referring to osteoporosis, you have to identify the most accredited source; in this case, the WHO. Please update your abstract, introduction and methods (Assessment of Bone Mineral Density) sections with this reference.

The diagnosis and classification of osteoporosis are based on BMD, which is reported as a T score and/or fracture history (source: World Health Organ Tech Rep Ser. 1994;843:1-129)

In methods section (Assessment of Bone Mineral Density), you report to have used "Achilles InSight ultrasound device (Achilles InSight, GE, USA)" (line 94-95). Please provide research studies that validate this method and its comparison with the Bone densitometry, that's the gold standard for diagnosing osteoporosis, based on dual-energy x-ray absorptiometry (DXA or DEXA) or Quantitative computed tomography (QCT).

You should consider this step as limiting stage in this review process. If there is no scientific evidence to support the methodology used and which ratify an equivalence with the gold-standard technique, the methodological premises are to be considered invalid and, consequently, the whole study.

The unique reference you cite is dated back to 2010

Jin, N.; Lin, S.; Zhang, Y.; Chen, F., Assess the discrimination of Achilles InSight calcaneus quantitative ultrasound device for osteoporosis in Chinese women: compared with dual energy X-ray absorptiometry measurements. Eur J Radiol 2010, 76, (2), 265-8.

and did not validate the method.

I will continue the review only if you produce what is requested.

Author Response

Dear Authors,

despite the novelty and scientific soundness of your research, many issues are hidden in the overall sufficient contribution proposal.

  1. First of all, when referring to osteoporosis, you have to identify the most accredited source; in this case, the WHO. Please update your abstract, introduction and methods (Assessment of Bone Mineral Density) sections with this reference. The diagnosis and classification of osteoporosis are based on BMD, which is reported as a T score and/or fracture history (source: World Health Organ Tech Rep Ser. 1994;843:1-129) 
  2. Ans: Thank you for your correction. We have changed the reference 1.
  3. In methods section (Assessment of Bone Mineral Density), you report to have used "Achilles InSight ultrasound device (Achilles InSight, GE, USA)" (line 94-95). Please provide research studies that validate this method and its comparison with the Bone densitometry, that's the gold standard for diagnosing osteoporosis, based on dual-energy x-ray absorptiometry (DXA or DEXA) or Quantitative computed tomography (QCT). You should consider this step as limiting stage in this review process. If there is no scientific evidence to support the methodology used and which ratify an equivalence with the gold-standard technique, the methodological premises are to be considered invalid and, consequently, the whole study. The unique reference you cite is dated back to 2010 Jin, N.; Lin, S.; Zhang, Y.; Chen, F., Assess the discrimination of Achilles InSight calcaneus quantitative ultrasound device for osteoporosis in Chinese women: compared with dual energy X-ray absorptiometry measurements. Eur J Radiol 2010, 76, (2), 265-8. and did not validate the method. I will continue the review only if you produce what is requested.
  4. Ans: Thank you for your comments. We have added one paragraph in the Discussion.
  • In this study, osteoporosis was confirmed using an Achilles InSight ultrasound device instead of dual energy X-ray absorptiometry (DXA). The Achilles InSight has been shown to be able to identify osteoporosis defined by axial BMD using DXA in Chinese women [53]. The QUS t-score at the left heel was positively correlated with the DXA t-score at the right heel (r = 0.90, p < 0.001) in eighty women, aged 53–73 years, with osteoporosis and/or fractures were followed repeatedly during 7 years [54]. Besides, QUS may be an improved predictor of fractures in comparison with DXA [55,56,57,58]. Although, the most widely validated technique to measure BMD is DXA, and diagnostic criteria based on the T-score for BMD are a recommended entry criterion for the development of pharmaceutical interventions in osteoporosis. However, quantitative ultrasound has several advantages over DXA, including that radiation is not required, the low cost and portability. (Line 312-322)

Reviewer 3 Report

  This is an interesting manuscript which has implications on causes of changes in bone mineral density (BMD) and the various disease associated with changes.  Of particular interest is he demonstration that ozone promotes increased BMD, while NOx and SO2 cause reduced BMD.  Curiously, while the authors state that the ozone results is their most important observation they fail to even mention it in their concluding summary.  This should be corrected. 

  There are some issues with tables and figures.  Table 2 adds nothing to the study and can be deleted.  At least this reviewer finds the 3-D figures impossible to understand in a 2-D image.   Figure 4 has important information but must be dramatically changed.  It is mostly blank space, and the legends are too small to see.  

In addition to the very interesting ozone result, the effects of CO, NOx and SO2 are also extremely important, given how common air pollution with elevated CO, NOx and SO2 occurs in many urban areas. 

Author Response

  This is an interesting manuscript which has implications on causes of changes in bone mineral density (BMD) and the various diseases associated with changes. Of particular interest is he demonstration that ozone promotes increased BMD, while NOx and SO2 cause reduced BMD.

  1. Curiously, while the authors state that the ozone results is their most important observation they fail to even mention it in their concluding summary. This should be corrected.
  2. Ans: Thank you for your comments. We have added in the summary.
  • In summary, we found that exposure to air pollutants such as CO, SO2, NO, NO2 and NOx was associated with osteoporosis, and O3 may be a protective factor against osteoporosis. (Line 338-339)
  •  
  1. There are some issues with tables and figures. Table 2 adds nothing to the study and can be deleted. At least this reviewer finds the 3-D figures impossible to understand in a 2-D image. Figure 4 has important information but must be dramatically changed. It is mostly blank space, and the legends are too small to see.
  2. Ans: Thank you for your comments. We have deleted Table 2. Regarding hard to understand the 3-D figures, we try to make the description of the 3-D figures clear. It is easier to look at 2 variables at a level. We also try to use photoimpact to make high Figure 4 pixel and larger font.
  • This was a combined analysis trying to model air pollution effect to BMD T-score based on levels of CO and NOX (Figure 2; p for interaction = 0.001, see Table 4) or levels of NO2 and SO2 (Figure 3; p for interaction = 0.004, see Table 4). (Line 193-195)
  •  
  1. In addition to the very interesting ozone result, the effects of CO, NOx and SO2 are also extremely important, given how common air pollution with elevated CO, NOx and SO2 occurs in many urban areas.
  2. Ans: Thank for your comments. We think so, woo. Despite the research limitations, our findings highlight the importance of air pollution on osteoporosis.

Reviewer 4 Report

It is an interesting topic to work on. The authors have analyzed properly the different issues concerning osteoporosis, ambient contamination, and gender.

The introduction is too short and it can be improved. The article by Prada et al. Environ Res June 185; 109465 published on the cellular and molecular mechanism on air pollution and bone damage should be reviewed and added by the authors since it provides a mechanism supporting the results presented. 

Minor details in English language should be corrected

Author Response

It is an interesting topic to work on. The authors have analyzed properly the different issues concerning osteoporosis, ambient contamination, and gender.

  1. The introduction is too short and it can be improved. The article by Prada et al. Environ Res June 185; 109465 published on the cellular and molecular mechanism on air pollution and bone damage should be reviewed and added by the authors since it provides a mechanism supporting the results presented.
  2. Ans: Thank you for your suggestion. We have added the issue in the Introduction.
  • According to recent review article, potential mechanisms behind the association between outdoor air pollution, especially particulate matter, and bone damage included 1) several different atmospheric pollutants can induce low-grade systemic inflammation, which affects bone metabolism through a specific effect of cytokines such as tumor necrosis factor α (TNF-α), interleukin (IL)-1β, IL-6, and IL-17 on osteoblast and osteoclast differentiation and function; 2) some pollutants, particularly certain gas and metal compounds, can cause oxidative damage in the airway and bone cells; 3) different groups of pollutants can act as endocrine disruptors when binding to the receptors in bone cells, changing their functioning; and 4) air pollution can directly and indirectly cause vitamin D deficiency [25]. (Line 61-69)
  •  
  1. Minor details in English language should be corrected.
  2. Ans: We have provided the English editing certificate as below.

Round 2

Reviewer 2 Report

Not bad, but please specify that further studies are needed to assess eventually the correlation between Achilles InSight ultrasound device (Achilles InSight, GE, USA) and the gold standard technique for diagnosing osteoporosis, based on dual-energy x-ray absorptiometry (DXA or DEXA) or Quantitative computed tomography (QCT) and consider it into limitations section.

Author Response

Not bad, but please specify that further studies are needed to assess eventually the correlation between Achilles InSight ultrasound device (Achilles InSight, GE, USA) and the gold standard technique for diagnosing osteoporosis, based on dual-energy x-ray absorptiometry (DXA or DEXA) or Quantitative computed tomography (QCT) and consider it into limitations section.

Ans: Thank you for your suggestion. We have added the issue in the limitation.

  • Third, further studies are needed to assess eventually the correlation between QUS and the gold standard technique for diagnosing osteoporosis, based on DXA or Quantitative computed tomography (QCT). (Line 324-326)
